# Diabetes Mellitus and Pneumococcal Pneumonia

**DOI:** 10.3390/diagnostics14080859

**Published:** 2024-04-22

**Authors:** Catia Cilloniz, Antoni Torres

**Affiliations:** 1Institut d’Investigacions Biomèdiques August Pi i Sunyer (IDIBAPS), University of Barcelona, 08036 Barcelona, Spain; cilloniz@recerca.clinic.cat; 2Centro de Investigación Biomédica en Red de Enfermedades Respiratorias (CIBERES), 28029 Madrid, Spain; 3Faculty of Health Sciences, Continental University, Huancayo 12001, Peru; 4Pulmonary Department, Hospital Clinic of Barcelona, C/Villarroel 170, 08036 Barcelona, Spain

**Keywords:** pneumonia, diabetes mellitus, pneumococcal pneumonia, pneumococcus

## Abstract

Currently, there are more than 500 million people suffering from diabetes around the world. People aged 65 years or older are the most affected by this disease, and it is estimated that approximately 96% of diabetes cases worldwide are type 2 diabetes. People with diabetes mellitus are at an increased risk of infections such as pneumonia, due to a series of factors that may contribute to immune dysfunction, including hyperglycemia, inhibition of neutrophil chemotaxis, impaired cytokine production, phagocytic cell dysfunction, altered T cell-mediated immune responses and the co-existence of chronic comorbidities. Rates of infection, hospitalization and mortality in diabetic patients are reported to be higher than in the general population. Research into the risk of infectious diseases such as pneumonia in these patients is very important because it will help improve their management and treatment.

## 1. Introduction

Community-acquired pneumonia (CAP) is a major global health problem associated with high rates of short- and long-term morbidity and mortality in all age groups, and also with high healthcare costs [1,2,3,4]. CAP can occur at any age, but its incidence and mortality risk increase with age and with the presence of chronic comorbidities (Figure 1, [5]). 

These comorbidities include respiratory diseases such as chronic obstructive pulmonary disease (COPD), cardiovascular disease, diabetes mellitus (DM), and chronic liver disease. Other populations at risk include those with HIV infections and other forms of immunosuppressive conditions, as well as people with chronic kidney disease and splenectomized patients [6].

*Streptococcus pneumoniae* (pneumococcus) remains the most common pathogenic cause of CAP [7]. Pneumococcus was reported to be one of the six leading pathogens for deaths associated with resistance; together, these pathogens were responsible for 929,000 deaths attributable to antimicrobial resistance worldwide [8]. A high burden of pneumococcal CAP has been reported by studies around the world [9,10]. Cases of severe pneumococcal CAP are associated with a higher risk of bacteremia, cardiac complications, sepsis, and septic shock, especially in people at increased risk such as those with diabetes mellitus [11,12,13,14,15,16,17]. Factors that may contribute to immune dysfunction in people with diabetes include hyperglycemia, inhibition of neutrophil chemotaxis, impaired cytokine production, phagocytic cell dysfunction, and altered T cell-mediated immune responses [18,19,20,21,22,23]. The International Diabetes Federation reported that 537 million adults were living with diabetes worldwide in 2021, and predicted that this figure would rise to 643 million by 2023. These numbers show the huge potential impact of pneumonia on patients with diabetes mellitus. Knowing the risk factors and clinical presentation of pneumococcal CAP in diabetic patients is important, for several reasons. First, diabetic patients are at an increased risk of contracting pneumonia due to immune dysfunction and the co-existence of underlying comorbidities. Second, pneumonia in diabetic patients often has severe outcomes and yields increased mortality rates. Third, pneumonia is a preventable disease, and so identifying the related risk factors would help the attempt to prevent it. The aim of this article is to review the epidemiology, clinical presentation, microbial etiology, and management of pneumococcal CAP in patients with diabetes mellitus.

## 2. Type 1 and Type 2 Diabetes

According to the American Diabetes Association (ADA), type 1 diabetes mellitus is defined as an autoimmune β-cell destruction, which leads to absolute insulin deficiency, while type 2 diabetes mellitus is defined as a progressive loss of β-cell insulin secretion mainly related to insulin resistance [24]. Hyperglycemia is a common feature of both types. The increased risk of infections in individuals with diabetes is related to the impact of hyperglycemia on our immune system, which affects, for example, the bactericidal functions of neutrophils, cellular immunity and complement activation [25]. The acute systemic inflammation triggered by pneumonia impairs the vascular wall and the endothelial permeability, which contributes to its pathophysiology [26].

## 3. Risk Factors for Pneumonia and Pneumococcal Pneumonia

People with diabetes mellitus are at an increased risk of pneumonia [27]. A population-based case–control study published a decade ago, which included data from 34,239 hospitalized CAP patients and 342,390 control patients, reported that the relative risk (RR) for pneumonia-related hospitalization in diabetic patients aged <40 years compared to those without diabetes was 3.2. The study also reported that the RR decreased with age, falling to 1.1 in patients aged ≥80 years [28]. Interestingly, patients with type 1 diabetes were associated with a 4.4-fold increased risk of pneumonia, while those with type 2 diabetes were associated with a 1.2-fold increased risk. The increased risk of pneumonia was associated with poor long-term glycemic control and longer diabetes duration [28]. Similarly, a retrospective, longitudinal cohort study including data from 77,637 patients with diabetes reported that high levels of A1C were associated with an increased risk of pneumonia in these patients [12]. These reports were also confirmed in a review study of the impact of diabetes in pneumococcal and invasive pneumococcal disease (IPD). The authors reported an increased risk of up to 1.4 for pneumococcal pneumonia and a risk ranging from 1.4 to 4.6 for IPD [29]. A recently published retrospective study reported that among adults aged 18 to 49 years, 50 to 64 years, and ≥65 years, the rates of pneumococcal pneumonia per 100,000 patients/year were 330.2, 572.2, and 1180.1, respectively. For the three groups, the rate ratios of adults with diabetes mellitus versus healthy counterparts were 2.9 (95% CI, 2.7–3.1), 3.0 (95% CI, 2.9–3.1), and 2.7 (95% CI, 2.6–2.7) (Figure 2, [30]).

In a retrospective study from Spain [31] of pulmonary complications in adult patients with pneumococcal CAP, the authors reported that 38% of the study population (n = 626) had the following pulmonary complications: pleural effusion (52%), empyema (8%) and multilobar infiltration (64%). Patients with pulmonary complications presented higher rates of ICU admission, higher rates of shock and longer hospital stay, but similar 30-day mortality. In the multivariate analyses, the authors reported that the presence of chronic liver disease, admission C-reactive protein level ≥18 mg/dL and creatine level >1.5 mg/dL were risk factors for pulmonary complications. In the study there were no differences in the serotype distribution between complicated and uncomplicated pneumonia [31].

Approximately 90% of pneumococcal cases are non-invasive pneumonia, and 10% are invasive pneumonia, which presents with bacteremia and empyema [32,33]. Another retrospective study from Spain [34], which analyzed data from 1948 adult patients with CAP between 2016 and 2020, reported that 85.1% had non-invasive pneumococcal CAP and 14.9% had the invasive form. Interestingly, the rate of non-invasive CAP was higher in adults aged 65 years and over (89%). In general, 36.3% of the cases of non-invasive pneumococcal pneumonia were caused by serotypes that are targeted by the pneumococcal conjugate vaccine PCV13. A similar proportion (38.2%) was reported in patients aged 65 years and over [34].

Finally, another retrospective study from India that analyzed data from 280 adult patients with non-invasive pneumococcal pneumonia reported that smoking, diabetes, and lung disease were the significant risk factors associated with non-invasive pneumococcal CAP [35].

In the case of invasive pneumococcal pneumonia, bacteremia is present in approximately 30% of patients with pneumococcal CAP. In a recent retrospective observational study that included data from 1783 patients with pneumococcal CAP, 33% of the patients presented bacteremia. In that study, nine factors were associated with the risk of bacteremia: no influenza vaccination in the last year, no pneumococcal vaccination in the last five years, blood urea nitrogen (BUN) level ≥30 mg/dL, sodium level <130 mmol/L, lymphocyte count <800/µL, C-reactive protein level ≥200 mg/L, respiratory failure, pleural effusion, and no previous antibiotic use before admission. These results are consistent with those of a previous study that identified high C-reactive protein levels (≥20 mg·dL^−1^), pleural effusion, and multilobar involvement to be independently associated with bacteremic CAP [36]. A multicenter retrospective study describing the evolution and distribution of CAP serotypes in Spain reported that 15% of patients with pneumococcal CAP presented invasive/bacteremic pneumonia, with the most frequently identified serotypes being serotype 8 (41%) and serotype 3 (14%) [34]. Interestingly, a retrospective observational study on the time to positivity (TTP) of blood cultures reported that a TTP less than 9.2 h in patients with bacteremic pneumococcal CAP was independently associated with more severe disease, characterized by a higher C-reactive protein level, worse oxygenation, and more pulmonary complications [37].

Pleural effusion and empyema are reported in 2% to 3% of CAP cases. Empyema is associated with higher morbidity and mortality [38,39,40]. A retrospective study from Saudi Arabia reported that of the 24 empyema cases included, 28% required ICU admission and 12% developed sepsis. The in-hospital and 90-day mortality rates reported in the study were 8% and 13%, respectively [41].

## 4. Why Do Diabetic Patients Have a Higher Risk of Pneumonia?

Individuals with diabetes have a dysregulated immune system and impaired defense mechanisms against microorganisms (Figure 3). 

The mechanisms that have been proposed to increase the risk of infection and the severity of pneumonia in diabetes are related to the effects of hyperglycemia, which decreases neutrophil degranulation, impairs complement activation, and disrupts phagocytosis [23,42].

Other factors include the chronic state of low-grade inflammation typical of individuals with diabetes, phagocytic and antibacterial dysfunction, unregulated inflammatory responses to viral infections and decreased viral clearance, and the presence of comorbidities such as cardiovascular disease [43,44,45,46,47,48]. Chronic hyperglycemia also affects innate and adaptative immunity [49]. During an infection, macrophages increase their phagocytic ability and generate reactive oxygen species (ROS), reactive nitrogen species (RNS), pro-inflammatory cytokines (IL-1β and IL-6), and anti-inflammatory cytokines (IL-10) to fight off the pathogens. This is part of the innate immune response that is affected by hyperglycemia in diabetic patients. Furthermore, there is a delay in the activation of T cell-mediated responses in diabetic patients, which increases the risk of infection [43,45,46].

Comorbidities such as cardiovascular disease, neurological disease, and chronic renal disease [50,51,52,53] are frequent in diabetic patients. Importantly, 90% of individuals with type 2 diabetes are at an increased risk of being overweight or obese [54]. These comorbidities increase the risk of infections such as pneumonia.

It is important to mention that during the COVID-19 pandemic diabetic patients were twice as likely to present severe infection as non-diabetics, and three times as likely to require ICU admission; they also presented higher mortality rates [18,55,56,57,58,59,60]. The association of diabetes mellitus and outcome severity was also reported in cases of severe acute respiratory syndrome (SARS), Middle East respiratory syndrome (MERS) and influenza virus AH1N1 [61,62,63,64,65].

## 5. Prognosis of Pneumonia in Diabetic Patients

Patients with diabetes and pneumococcal CAP present a high rate of complications and mortality. A retrospective study including data from 215 patients with diabetes and CAP reported that 43% developed complications, with respiratory failure being the most common [66]. The study also noted that the in-hospital mortality in these patients with complications was 30%. Another retrospective study, which investigated the outcomes of 1262 patients with type 2 diabetes and severe CAP and 2425 patients with severe CAP but without type 2 diabetes [67], reported a longer ICU stay (13 vs. 12 days, *p* = 0.016), higher 30-day mortality (26% vs. 23%, *p* = 0.046), higher ICU mortality (31% vs. 27%, *p* = 0.005), and higher in-hospital mortality (35% vs. 31%, *p* = 0.009) in patients with type 2 diabetes. In these patients, increased numbers of comorbidities and diabetes-related complications, elevated C-reactive protein levels, elevated neutrophil to lymphocyte ratio levels, elevated levels of brain natriuretic peptide and blood lactate, and decreased blood pressure on admission were independent risk factors for in-hospital mortality.

Other important complications in pneumonia include related cardiac events, which are reported in between 10% and 30% of cases of CAP [13,14,68]. Patients with severe pneumococcal CAP have an increased risk of a cardiac complication (e.g., myocardial infarction, arrhythmia, and/or congestive heart failure) [69]. Results from a prospective multicenter study including hospitalized CAP patients reported that age, smoking, chronic heart disease, initial severity of infection, and pneumococcal infections were risk factors for both early (30-day follow-up) and late (one-year follow-up) cardiovascular complications [69]. In that study, 21% (n = 423) of the population had diabetes mellitus, among whom 28% (n = 56) and 33% (n = 40) had early and late cardiac complications, respectively [67]. Pneumolysin (Ply), a pneumococcal toxin, and the systemic inflammation in cases of severe pneumococcal pneumonia may explain these cardiac complications [10,70,71], together with the micro- and macrovascular complications developed by patients with diabetes mellitus during their lifetime [72]. An interesting animal experiment on pneumococcal pneumonia and cardiac events reported that high-grade bacteremia is a requisite for cardiac damage, and observed that the mechanism of cardiac damage varied between strains [73]. It has been reported that the measurement of pneumococcal DNA load may identify patients with higher risk of severe complications, including those with cardiac events [74,75].

Finally, co-infection is a complication that is increasingly recognized in patients with CAP. In a retrospective observational study of CAP patients with severe pneumonia, 11% of the study population presented polymicrobial CAP. In that study, the most frequent combination of pathogens was *S. pneumoniae* and influenza virus [76]. The in-hospital mortality in patients with polymicrobial CAP was 21%, and it was 11% in patients with monomicrobial CAP. These results are similar to those of other studies. There have also been reports of bacterial co-infection between *S. pneumoniae* and influenza virus AH1N1 during the pandemic in 2009, and in cases of COVID-19 pneumonia [77,78,79].

*S. pneumoniae* colonize the upper respiratory tract, and the initiation of infection is preceded by a viral infection. An interesting study showed that influenza virus increases the accessibility of host sialylated mucins in the upper respiratory, which promotes the growth and spread of *S. pneumoniae* [44].

A retrospective observational study, analyzing the data from 6403 CAP patients from the period 1997 to 2016, reported that the mortality rate of pneumococcal CAP had not fallen in the last two decades. The study also found diabetes mellitus to be an independent risk factor for 30-day mortality together with age ≥65 years, an SOFA score of ≥5, and a requirement for mechanical ventilation (non-invasive and invasive) [80]. The study by Wagenvoort et al. [81] demonstrated a significant increase in long-term mortality among patients with pneumococcal pneumonia. In the study, approximately 40% of all patients who survived the first 30 days after presentation of pneumococcal pneumonia died within the following five years. PCV7 serotype disease was independently associated with long-term mortality in patients with invasive pneumococcal pneumonia who survived 30 days after the infection. The damage during acute pneumococcal pneumonia may explain this long-term mortality among patients with chronic conditions such as diabetes mellitus and pneumococcal pneumonia [82].

Significantly, metformin use in type 2 diabetes is associated with a low risk of pneumonia [83,84,85]. A recent study analyzing data from 34,759 elderly (≥65 years) patients with type 2 diabetes and CAP reported that the 30-day (RR: 0.86; 95% CI: 0.78–0.95) and 90-day (RR: 0.85; 95% CI: 0.79–0.92) mortality rates after propensity score matching were significantly lower in the patients who used metformin than in those who did not [85].

## 6. Prevention

### 6.1. Diabetes Control

Lifestyle interventions with or without medication to prevent type 2 diabetes in high-risk populations are an important measure for reducing the risk of complications and, in turn, the risk of pneumonia [52,86,87,88].

In 2021, the Lancet Commission on Diabetes [52] recommended a drive to change policy responses globally and to close the gaps in diabetes prevention, care, and professional knowledge. A key message was the need to ensure access to insulin, patient education, and tools for monitoring blood glucose concentrations that might prevent premature deaths. Another key message was the need to develop strategies (environmental, behavioral, and socioeconomic) for preventing type 2 diabetes, which require multilevel involvement across society and across populations.

An interesting example of managing type 2 diabetes was provided by a randomized controlled trial (RCT) that evaluated the effect of telehealth education on controlling the condition in 174 of the patients affected [89]. After 26 weeks of the program, there were statistically significant reductions in weight, body mass index, fasting blood glucose levels, 2 h postprandial blood glucose levels, and hemoglobin A1c when compared to the control group (*p* < 0.05). Significant reductions in systolic blood pressure and low-density lipoprotein-C levels (*p* < 0.05) were also observed in the group receiving the telehealth education. These results show the importance of telehealth education in controlling type 2 diabetes.

Metformin is the most commonly used antidiabetic medication worldwide, and has been shown to lower blood sugar levels and modulate the metabolism. A recent study investigated the risk of pneumonia in 49,012 propensity score-matched metformin users and non-users with type 2 diabetes [84]. The mean age of the sample was 57.4 (12.8) years, and the mean follow-up periods were 5.47 years and 5.15 years for metformin users and non-users, respectively. The study showed that metformin use was associated with a significantly lower risk of bacterial pneumonia, invasive mechanical ventilation, and death compared to non-use. Another retrospective study evaluated the association of prior metformin use with 30-day and 90-day mortality in older (≥65 years) diabetic patients hospitalized with pneumonia [85]. In that study, which included 34,759 patients, 20% received metformin. After propensity score matching, the 30-day (RR: 0.86; 95% CI: 0.78–0.95) and 90-day (RR: 0.85; 95% CI: 0.79–0.92) mortality rates were significantly lower for the metformin users. These results suggest that the prior use of metformin was associated with lower mortality in elderly diabetic patients hospitalized with pneumonia.

These examples of interventions show the importance of controlling type 2 diabetes in order to reduce the risk of infections in this population.

### 6.2. Vaccination

Pneumococcal and influenza vaccines provide protection against the two most frequent pathogenic causes of CAP. Recently, a respiratory syncytial virus vaccine was approved for use. Currently, there is one pneumococcal polysaccharide vaccine (PPV) and three pneumococcal conjugate vaccines (PCV). These vaccines are: the 23-valent PPV (targets the serotypes 1, 2, 3, 4, 5, 6B, 7F, 8, 9N, 9V, 10A, 11A, 12F, 14, 15B, 17F, 18C, 19A, 19F, 20, 22F, 23F, and 33F), the 13-valent PCV13 (covers the serotypes 1, 3, 4, 5, 6A, 6B, 7F, 9V, 14, 18C, 19A, 19F, and 23F), the 15-valent PCV15 (covers the serotypes 1, 3, 4, 5, 6A, 6B, 7F, 9V, 14, 18C, 19A, 19F, 22F, 23F, and 33F), and the 20-valent PCV20 (targets the serotypes 1, 3, 4, 5, 6A, 6B, 7F, 8, 9V, 10A, 11A, 12F, 14, 15B, 18C, 19A, 19F, 22F, 23F, and 33F) (Figure 4).

The recommendations of the Advisory Committee on Immunization Practices (ACIP) for pneumococcal vaccines are shown in Figure 5 and Figure 6. 

Individuals with diabetes mellitus are included in the groups that are at increased risk of pneumococcal disease (Figure 7).

The new ACIP recommendations [90] include the use of PCV20 or PCV15 alone or in series with PPV23 in all adults aged ≥19 years, including those with diabetes mellitus who have received a PCV or whose vaccination history is unknown.

The pneumococcal vaccine is the most important vaccine for the prevention of pneumonia in individuals with diabetes mellitus. Patients with diabetes mellitus are reported to have a higher risk of developing CAP (up to 1.4 times higher) and invasive pneumococcal disease (IPD) (1.4 to 5.9 times higher) than the general population. The risk of pneumococcal disease is higher among younger adults with diabetes than in their older peers. It is also important to mention that the comorbidities associated with diabetes mellitus can increase the risk of pneumococcal disease and that the persistent hyperglycemia in these individuals impairs the immune system and increases the risk of infection [29].

A randomized controlled trial study among elderly (≥65 years) individuals with comorbidities investigated the vaccine efficacy (VE) of PCV13 in patients with diabetes [91]. The authors reported that diabetes mellitus significantly modified VE, reaching a level of 89% (95% CI, 65.5–96.8) in immunocompetent elderly patients with diabetes mellitus compared with 25% (95% CI, −10.4 to 48.7) in non-diabetics. Despite the interest in this result, the authors did not seek to elucidate the underlying biological mechanism. In fact, little is known about the role of diabetes mellitus as a modifier of vaccine immunogenicity and efficacy [92].

The co-infection of pneumococcus with the influenza virus is common [44,93]. To avoid the risk of co-infection and severe infection (which in many cases is associated with poor outcomes), the use of the influenza vaccine is recommended. This vaccine reduces the risk of influenza-related pneumonia, which is a major complication in these infections [94,95].

Despite the importance of pneumococcal and influenza vaccination in patients with diabetes mellitus, an observational retrospective study from Spain found adherence to vaccination programs in this population to be low [96]: 55% in the case of vaccination for influenza, 18% for pneumococci, and 17% for hepatitis B. In that study the predictors of correct vaccination were: younger age, shorter diabetes duration, insulin pump treatment, better diabetes control and being a health professional [96]. Similar findings were reported in a cross-sectional survey from Saudi Arabia, where the prevalences of influenza and pneumococcal vaccination were 47% and 35%, respectively. In that study, the authors reported that older individuals, those who were unmarried and less educated, and those living with certain chronic conditions were less likely to have received vaccination [97]. Data from a cross-sectional multicenter study from Turkey (the TEMD vaccination study) showed that patients with diabetes (type 1 and type 2) had very low rates of influenza and pneumococcal vaccination [98]. Similar findings have been reported in other countries [99,100,101,102].

Annual influenza vaccination is recommended for individuals over the age of six months, healthcare workers, immunocompromised patients, and individuals with comorbidities.

## 7. Conclusions

Pneumococcal pneumonia is a common lung infection, associated with multiple complications and high mortality. The risk of pneumococcal pneumonia is higher in people with either type 1 or type 2 diabetes mellitus than in those without diabetes. This increased susceptibility to pneumonia is in great part due to the chronic inflammation mediated by the hyperglycemic state, which disrupts a wide range of immune responses. The increasing global burden of diabetes mellitus makes it necessary to take preventive measures to protect patients from pneumonia and its serious consequences. This highlights the important role of vaccines, such as pneumococcal and influenza vaccines, in protecting these patients.

## Figures and Tables

**Figure 1 diagnostics-14-00859-f001:**
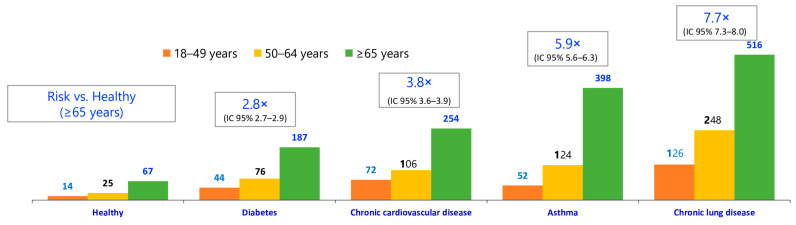
The incidence of pnemococcal pneumonia increases with age, rate of comorbidities, Incidence rate (100,000 people/year) and rate ratio in healthy and at-risk adults.

**Figure 2 diagnostics-14-00859-f002:**
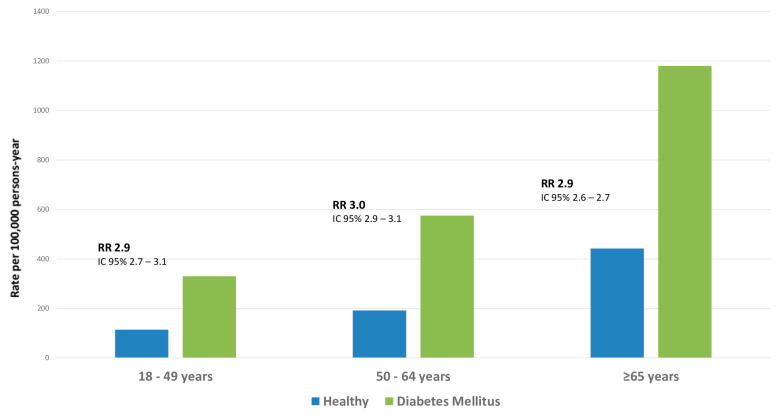
Rate of pneumococcal pneumonia by age and presence of diabetes mellitus.

**Figure 3 diagnostics-14-00859-f003:**
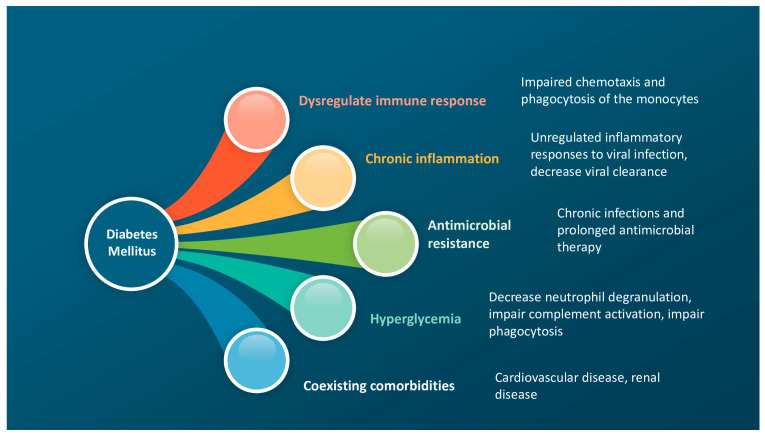
Diabetes mellitus and the risk of pneumonia and poor outcomes.

**Figure 4 diagnostics-14-00859-f004:**
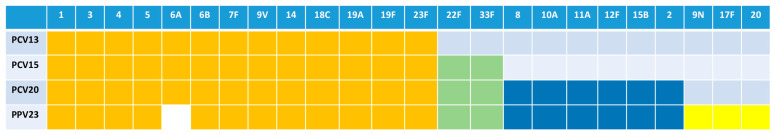
Pneumococcal vaccines.

**Figure 5 diagnostics-14-00859-f005:**
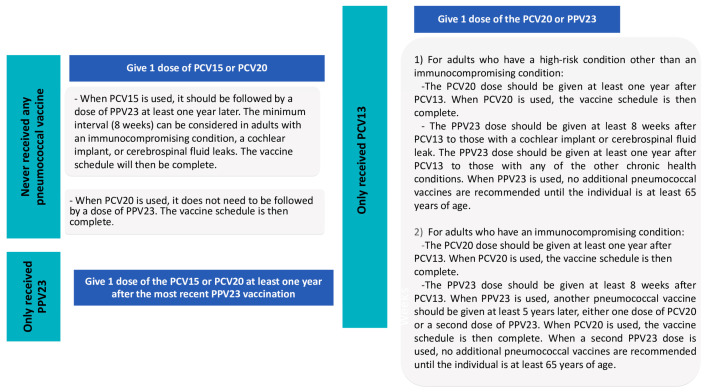
ACIP recommendations for pneumococcal vaccines: Part 1.

**Figure 6 diagnostics-14-00859-f006:**
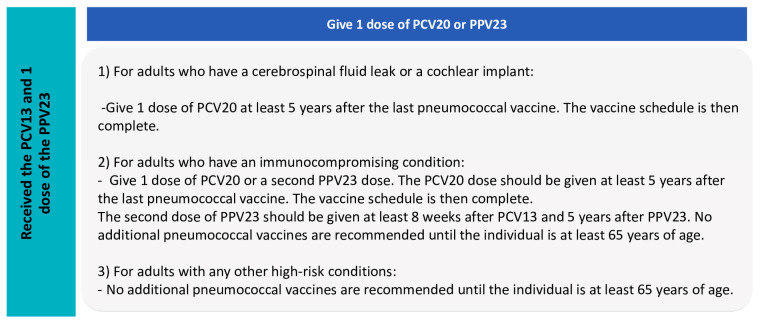
ACIP recommendations for pneumococcal vaccines, Part 2.

**Figure 7 diagnostics-14-00859-f007:**
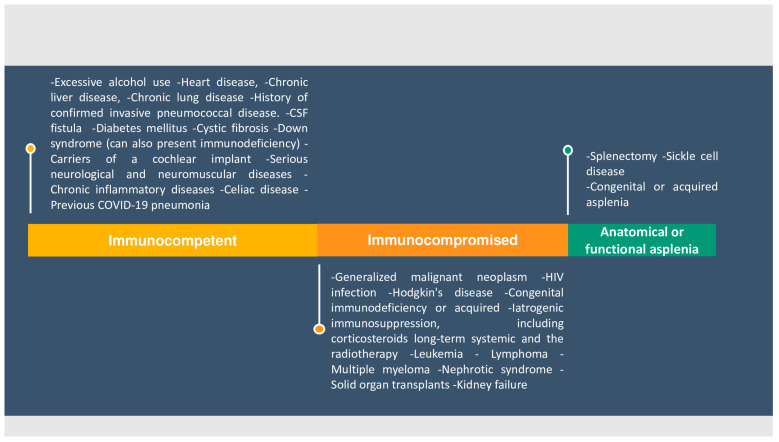
Conditions and risk factors where pneumococcal vaccination is indicated.

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
