# Peer review of "Diabetes Mellitus and Pneumococcal Pneumonia"

_diagnostics, 2024, doi:10.3390/diagnostics14080859_

Round 1
Reviewer 1 Report
Comments and Suggestions for Authors
This is an interesting review that analyzes the relationship between diabetes mellitus and pneumococcal pneumonia.
The review is very well structured and gives a complete review of the current situation, for which I would like to congratulate the authors.
I would just like to make a few comments:
- In the abstract world appears twice in the same sentence, one should be removed.
- Figures 1 and 2 do not read well (at least in the review version I received) and seem to be taken from an article. Do you have the authorization to use them?
- In the second section they could consider changing the title, since for several paragraphs they talk about risk factors that are not DM.
- Tables 1 and 2 could be changed to make them more visual.
Author Response
Reviewer 1
This is an interesting review that analyzes the relationship between diabetes mellitus and pneumococcal pneumonia. The review is very well structured and gives a complete review of the current situation, for which I would like to congratulate the authors. I would just like to make a few comments:
- In the abstract world appears twice in the same sentence, one should be removed.
Answer: Thank you for your comments. We have corrected the mistake you mention.
- Figures 1 and 2 do not read well (at least in the review version I received) and seem to be taken from an article. Do you have the authorization to use them?
Answer: Thank you for your comment. We created these figures; however, we noticed that the transformation from jpg to pdf affected the quality, and we have now modified them accordingly.
- In the second section they could consider changing the title, since for several paragraphs they talk about risk factors that are not DM.
Answer: Thank you for your comment. We have changed the title in accordance with your suggestion (p. 4, l. 99)
- Tables 1 and 2 could be changed to make them more visual.
Answer: Thank you for your comment. We have modified our tables accordingly.
Reviewer 2 Report
Comments and Suggestions for Authors
In the peer-reviewed review, the authors consider diabetes as a risk for infections like pneumonia. This report is descriptive. The authors do not discuss immunologic and biochemical disorders in type 1 and type 2 diabetes. Although these disorders are the risk of bacterial infection. I realize that it is impossible to cover everything in a short review, but it would be worth discussing the abnormal biochemical processes in these patients in the chapter on diabetes as a risk for pneumonia. In diabetes, the permeability of the vascular wall is impaired, which is a favorable factor for the spread of bacteria. I think this could also be mentioned.
The figures in the publication are of very poor quality, they are just unreadable. The description of the figures is practically absent. The quality of figures does not meet the journal's rules.
Author Response
Reviewer 2
In the peer-reviewed review, the authors consider diabetes as a risk for infections like pneumonia. This report is descriptive. The authors do not discuss immunologic and biochemical disorders in type 1 and type 2 diabetes. Although these disorders are the risk of bacterial infection. I realize that it is impossible to cover everything in a short review, but it would be worth discussing the abnormal biochemical processes in these patients in the chapter on diabetes as a risk for pneumonia. In diabetes, the permeability of the vascular wall is impaired, which is a favorable factor for the spread of bacteria. I think this could also be mentioned.
Answer: Thank you for your comment. We have now added a paragraph on the immunological and biochemical disorders in type 1 and type 2 diabetes (p. 4, l. 99 to 109
The figures in the publication are of very poor quality, they are just unreadable. The description of the figures is practically absent. The quality of figures does not meet the journal's rules.
Answer: Thank you for your comment. We noticed that the transformation from jpg to pdf affected the quality of the figures; we have now improved them.